# Effects of Coated Cysteamine on Oxidative Stress and Inflammation in Weaned Pigs

**DOI:** 10.3390/ani11082217

**Published:** 2021-07-27

**Authors:** Shanshan Wang, Miaomiao Bai, Kang Xu, Yirui Shao, Zhe Yang, Xia Xiong, Ruilin Huang, Yao Li, Hongnan Liu

**Affiliations:** 1Hunan Provincial Key Laboratory of Animal Nutritional Physiology and Metabolic Process, National Engineering Laboratory for Pollution Control and Waste Utilization in Livestock and Poultry Production, Key Laboratory of Agro-Ecological Processes in Subtropical Region, Hunan Provincial Engineering Research Center for Healthy Livestock and Poultry Production, Scientific Observing and Experimental Station of Animal Nutrition and Feed Science in South-Central, Ministry of Agriculture, Institute of Subtropical Agriculture, Chinese Academy of Sciences, Changsha 410125, China; wss19970630@163.com (S.W.); miaomiaobai1115@126.com (M.B.); xukang2020@163.com (K.X.); tonyshao1006@foxmail.com (Y.S.); xx@isa.ac.cn (X.X.); huangrl@isa.ac.cn (R.H.); 2Institute of Animal Nutrition, Northeast Agricultural University, Harbin 150030, China; 3College of Animal Science and Technology, Hunan Agricultural University, Changsha 410128, China; promise.yangzhe@foxmail.com

**Keywords:** cysteamine, inflammatory cytokines, intestinal barrier, oxidative stress

## Abstract

**Simple Summary:**

Oxidative stress and inflammation are common problems in livestock and poultry production, and have a serious impact on animal welfare and profitability. Finding an effective way to deal with oxidative stress and inflammation is an urgent problem for the modern pig industry. Cysteamine has been shown to play a crucial role in the growth performance, digestive metabolism, immune function, and fecundity of animals. In this study, we found that dietary supplementation of coated cysteamine could enhance the level of immunoglobulin and the expression of intestinal immune factors in the weaned pigs induced by diquat. It is proved that dietary supplementation with coated cysteamine can improve the intestine barrier function and immune function.

**Abstract:**

This study aimed to explore the effects of dietary coated cysteamine on oxidative stress and inflammation in diquat-induced weaning pigs. Twenty-four pigs were randomly assigned to three dietary groups with eight replicates: the control (fed base diet), diquat (fed base diet), and coated cysteamine + diquat groups (fed 80 mg/kg cysteamine). The experiment was conducted for 21 d, and consisted of a pre-starter period (14 d) and a starter period (7 d). Coated cysteamine treatment significantly increased (*p* < 0.05) the final weight and average daily gain (ADG) in pigs. The contents of alkaline phosphatase (ALP), immunoglobulin G (IgG), serine (Ser), and isoleucine (Ile) were elevated (*p* < 0.05) while the contents of albumin (ALB) and aspartic acid (Asp) were reduced (*p* < 0.05) in the serum after coated cysteamine supplementation. Coated cysteamine supplementation resulted in greater (*p* < 0.05) serum superoxide dismutase (SOD) activity, the expression of interleukin-10 (IL-10) mRNA in the colon, and the CuSOD mRNA expression in the jejunum (*p* < 0.05) and colon (*p* = 0.073). Coated cysteamine supplementation showed an increasing trend in villus height (*p* = 0.060), villus height/crypt depth (V/C) (*p* = 0.056), the expression levels of zonula occludens-1 (ZO-1) mRNA (*p* = 0.061), and Occludin mRNA (*p* = 0.074) in the jejunum. In summary, dietary supplementation with coated cysteamine improves the intestinal barrier function of the jejunum by increasing the immunoglobulin content and the relative expression of intestinal immune factor mRNA in pigs while alleviating oxidative stress and inflammatory reactions caused by diquat.

## 1. Introduction

Oxidative stress is a frequent phenomenon that occurs in living organisms. Under normal circumstances, the oxidation and antioxidant systems in the body reach a balanced state [1], whereas the body reacts to oxidative stress [2] and produces excessive amounts of reactive oxygen species (ROS) when this balance is broken, which is likely to cause cytotoxicity [3]. ROS results from mitochondrial metabolism in eukaryotic cells, and plays an essential role in maintaining cell homeostasis and low-level regulation of signal transduction, gene expression, and receptor activation [4]. Several chronic diseases are associated with oxidative stress which can cause severe cell and tissue damage [5]. In the practical production process, oxidative stress reduces growth performance and immunity in pigs, which in turn causes various diseases and huge losses to the breeding industry [6]. Therefore, finding an effective method to alleviate or solve oxidative stress in pigs is a problem requiring an urgent solution.

Diquat, 1,1′-ethylene-2,2′-bipyridinium dibromide, generally exists as a dibromide salt. It is a non-selective, quick-acting bipyridine compound with biotical properties [7]. As diquat affects the performance and alimentation efficiency of animals, it is used extensively to induce oxidative stress [8]. Diquat has been proven to reduce the growth performance of animals, induce apoptosis and autophagy, reduce cell viability, reduce dopamine and antioxidant enzyme levels, generate a large amount of ROS, and destroy intestinal morphology and intestinal barrier function by inhibiting mucosal cell proliferation, and reducing intestinal permeability [9,10].

Cysteamine (Cys), also known as 2-mercaptoethylamine, is a new feed additive that is produced as a factor of the Coenzyme A pathway and is biologically derived from cysteine metabolism. Coenzyme A is synthesized from pantothenic acid and cysteine [11]; during its degradation process, pantetheine is formed and then hydrolyzed to cysteamine and pantothenic acid by pantetheinase [12]. Previous reports have confirmed that cysteamine can improve animal growth performance and carcass quality [13,14], increase the contents of glutathionase and superoxide dismutase (SOD) in cell tissues, reduce the content of malondialdehyde (MDA), and increase the production of gastrin and gastric acid. Thus, Cys exerts effects on oxidative stress and animal intestinal ulcer models [15,16]. Cys is easily degraded and rapidly oxidized in air, or generates disulfide cystamine in solution [17]. Therefore, enteric coating of cysteamine is an effective method to prevent it from dissolving in the acidic environment of the stomach [18]. Coated cysteamine is a feed stable cysteamine hydrochloride produced by advanced microencapsulation technology, adopting an advanced microcapsule coating technology and special coating wall materials with an encapsulation rate of 100%. The coated cysteamine has better fluidity and stability, better sustained-release performance, and better tolerance [19]. In addition, coating cysteamine can effectively avoid the loss caused in the process of processing and utilization and at the same time avoid causing animal gastrointestinal ulcers [12].

The purpose of our study was to explore the influence of coated cysteamine on oxidative stress and the inflammatory response in diquat-induced pigs models. We hypothesized that cysteamine coating has a moderating effect on oxidative stress and the inflammatory response in diquat-induced pigs by enhancing the antioxidant capacity and intestinal barrier function in pigs.

## 2. Materials and Methods

### 2.1. Animal Experiment Design

All animals used in this study were humanely managed according to the Chinese Guidelines for Animal Welfare. The experimental scheme was approved by the Animal Welfare Committee of the Institute of Subtropical Agriculture, Chinese Academy of Sciences (2019-6A). Twenty-four weaned pigs, aged 35 d (Duroc × Yorkshire), with a body weight of 9 ± 0.6 kg were enrolled in the trial. Pigs were fed a basal diet for seven days before the beginning the trials. All pigs were divided into three treatment groups with eight replicates at random: the control group, the diquat group, and the coated cysteamine + diquat group (Cys + diquat group), in which the control group and the diquat group were fed a basal diet, whereas the Cys + diquat group was fed an 80 mg/kg cysteamine diet. The trial lasted for 21 d. On the 14th day, the pigs were weighed and treated with diquat for 7d (85-00-7, >95.00%, Sigma-Aldrich, St. Louis, MO, USA). The diquat and Cys + diquat groups were intraperitoneally injected with diquat (10 mg/kg BW) to induce oxidative stress, while the control group was injected with the same amount of normal saline for 7 d. The experimental diet was designed to meet or exceed the nutrient requirements for weaned pigs (NRC 2012) (Table 1). Referring to Bai et al. [20], cysteamine was added to the feed in the form of coated cysteamine, which contains 27% cysteamine hydrochloride (Hangzhou King Techina Technology Co Ltd., Hangzhou, China). Before the test, the piggery was cleaned and disinfected thoroughly according to the piggery management process. During the test, the house was kept clean and dry, proper air circulation was ensured, and the house was cleaned every day. All piglets are housed in an environmentally healthy nursery facility with good heat preservation facilities and a mechanical ventilation system. The adjacent fields are separated by steel tubes to ensure that the piglets are not completely isolated from each other. The piggery is equipped with an automatic drinking water device, manual feeding is adopted and the trough is cleaned in time. The animals were allowed water and feed ad libitum and were fed in single pens throughout the study.

### 2.2. Sample Collection

On the 21st day, following a fast for 12 h before slaughtered, 10 mL blood samples were collected from the precaval vein with vacuum tubes and centrifuged at 3000 rpm for 5 min. The obtained upper serum was stored at −20 °C for future analysis. Moreover, jejunum and colon samples were collected and stored at −80 °C.

### 2.3. Growth Performance

All pigs were weighed on the first and the 21st day of the experiment, and the feed intake was calculated daily. The average daily gain (ADG), average daily feed intake (ADFI), and feed-to-gain ratio (F/G) were subsequently calculated.

### 2.4. Serum Biochemical Index

The levels of glucose (GLU), triglycerides (TG), cholesterol (CHO), total protein (TP), albumin (ALB), blood urea nitrogen (BUN), alkaline phosphatase (ALP), alanine aminotransferase (ALT) and aspartate aminotransferase (AST) were determined using an automatic biochemical analyzer (Cobas311, F. Hoffmann-La Roche Ltd., Basel, Switzerland). The kits used for the determination of the above indicators were purchased from Roche, Switzerland. Serum levels of immunoglobulin M (IgM, ab190537, Pig, Abcam, Wuhan, China), immunoglobulin G (IgG, KA2016, Pig, Abnova, Wuhan, China), and immunoglobulin A (IgA, ab190536, Pig, Abcam, Wuhan, China) in serum were determined by enzyme-linked immunosorbent assay. The operation steps strictly followed the manufacturer’s instructions.

### 2.5. Serum Amino Acids

Approximately 2 mL of blood was centrifuged for 5 min at 3000 rpm to separate the solids. The supernatant (1 mL) was added to 1 mL of 8% sulfosalicylic acid. The mixture was incubated for 15 min and centrifuged at 3000 rpm for 20 min. Amino acid concentrations were measured using an automatic amino acid analyzer (L-8800A; Hitachi, Tokyo, Japan).

### 2.6. Jejunal Morphology

After dissection, 1–2 cm of intestinal tissue from the middle jejunum of pigs was excised, digested, fixed in 4% paraformaldehyde solution and gradually dehydrated with an ethanol gradient of 75% → 85% → 95% → 100% → 100%. Samples were then cleaned with xylene, embedded in paraffin, processed into sections and stained with hematoxylin and eosin (HE). A positive fluorescence microscope (DM4000B, Leica Microsystems, Germany) was used to measure the villus height and crypt depth to subsequently calculate the villus height/crypt depth ratio.

### 2.7. Serum Antioxidant Indicators

One milliliter of serum was centrifuged at 3000 rpm for 15 min and the supernatant was drawn for the determination of antioxidant indices. The kits from Nanjing Jiancheng Bioengineering Institute (A001-3-2; A007-1-1; A015-1-2; A003-1-1; A044-1-1; Nanjing, China) were used for the detection of SOD, MDA, catalase (CAT), total antioxidant capacity (T-AOC) and myeloperoxidase (MPO).

### 2.8. Fluorescence Quantitative Detection of the Jejunum and Colon

RNA was extracted from jejunum and colon tissues using the Trizol kit (Invitrogen, Carlsbad, ON, Canada). RNA concentrations (ng/UL) and the A260/A280 ratio were recorded using a NanoDrop Spectrophotometer (NanoDrop, Wilmington, NC, USA). According to the study by Roy et al. [21], RNA was reverse transcribed into cDNA after passing the integrity test, and was used as a template for real-time quantitative polymerase chain reaction (PCR) detection with the Takara reverse transcription kit. The PCR was performed under the following conditions: 95 °C for 30 s, denaturation at 95 °C for 5 s, annealing at 51–60 °C for 30 s, and a total of 40 cycles. Finally, a 20 μL reaction system was used for fluorescence quantification. Quantitative gene and primer sequences are shown in Table 2.

### 2.9. Western Blotting Analysis

Relative protein levels of CuSOD, MnSOD, GPX1, GPX4, ZO-1, Claudin-1, Occludin, IL-2, IL-8, and IL-10 in the jejunum were determined using western blotting. Colon samples were collected and the protein expression of CuSOD, MnSOD, GPX1, GPX4, IL-2, IL-8 and IL-10 was determined [22]. The resultant signals were obtained using Quantity One software (Bio-Rad, Hercules, CA, USA). Primary antibodies were used as follows: CuSOD (1:50000; ab51254, Rabbit, Abcam, UK), MnSOD (1:1000; ab68155, Rabbit, Abcam, UK), GPX1 (1:1000; bs-3882R, Rabbit, Bioss, Beijing, China), GPX4 (1:1000; 14432-1-AP, Rabbit, Proteintech, Rosemont, IL, USA), ZO-1 (1:3000; 21773-1-AP, Rabbit, Proteintech, USA), Occludin (1:1000; ab167161, Rabbit, Abcam, UK), Claudin1 (1:500; ab15098, Rabbit, Abcam, UK), IL-2 (1:2000; ab92381, Rabbit, Abcam, UK), IL-8 (1:2000; ab110727, Rabbit, Abcam, UK), IL-10 (1:1000; 20850-1-AP, Rabbit, Proteintech, USA) and Actin (1:5000; 66009-1-Ig, Mouse, Proteintech, USA).

### 2.10. Statistical Analysis

The pigs were treated with the independent variables cysteamine and diquat, and explored their effects on various indicators. All data are shown as the mean ± standard error of mean (SEM) and were analyzed using one-way ANOVA (SPSS 21.0, SPSS Inc., Chicago, IL, USA). Duncan’s method was used for multiple comparisons. A value of *p* < 0.05 indicates statistical significance, and 0.05 < *p* < 0.1 indicates a decreasing or increasing trend.

## 3. Results

### 3.1. Growth Performance

Pigs’ weight increased during the test period (*p* < 0.05), and the control group had the highest final weight, followed by the Cys + diquat group and the lowest in the diquat group (Table 3). We found that there were no significant differences in ADG, ADFI and F/G among the three groups on 1–7 days and 8–14 days (*p* > 0.05). The ADG and ADFI in the diquat group were significantly lower than those in the control group on 15–21 days (*p* < 0.05), but there was no difference between the diquat group and the Cys + diquat group (*p* > 0.05).

### 3.2. Serum Biochemical Index

The control and Cys + diquat groups showed marked decreased (*p* < 0.05) in the content of ALB and GLU but increased (*p* < 0.05) levels of ALT and ALP compared to the diquat group (Table 4). The TP content in the Cys + diquat group was the lowest among the three group (*p* < 0.01). Serum IgA levels (*p* < 0.05) were lower in the diquat and Cys + diquat groups. Compared to the diquat group, the control and the Cys + diquat groups had higher (*p* < 0.05) serum IgG levels.

### 3.3. Serum Amino Acids

The results for the serum amino acids are shown in Table 5. The Cys + diquat group and the control group markedly increased (*p* < 0.05) the contents of Ser and Ile, and decreased (*p* < 0.05) the Asp content compared with the diquat group. The content of glycine (Gly) decreased (*p* < 0.05) in the Cys + diquat group compared to that in the control group.

### 3.4. Jejunal Morphology

As shown in Figure 1, the distribution of villus in the Cys + diquat group is tighter than that of the other two groups, and the shape of villus is more complete. As shown in Table 6, the diquat group showed a decline (*p* < 0.05) in villus height and V/C in the pigs jejunum compared with the control group. There were no significant changes among the Cys + diquat group and the other two groups.

### 3.5. Serum Antioxidant Index

As presented in Table 7, the Cys + diquat group showed increased SOD activity compared to the control and diquat group (*p* < 0.05). However, no significant difference in the serum antioxidant indices (*p* > 0.05) was observed between the diquat and control groups.

### 3.6. Relative Cytokine mRNA Levels in the Jejunum and Colon

The expression levels of related genes in the jejunum and colon are shown in Figure 2. Compared with the diquat group, coated cysteamine supplementation remarkably upregulated CuSOD mRNA expression (*p* < 0.05) and tended to increase (0.05 < *p* < 0.1) the relative expression of ZO-1 and Occludin mRNA in the pigs jejunum (Figure 2a). The mRNA levels of IL-4 in the Cys + diquat and diquat groups were lower (*p* < 0.05) than those in the control group, but did not differ between the Cys + diquat and diquat groups (*p* > 0.01). The expression level of IL-8 was higher (*p* < 0.05) in the diquat group than in the control group.

In the colon, the Cys + diquat group showed a remarkable increase (*p* < 0.05) in the expression of IL-10 mRNA and tended towards an increase (0.05 < *p* < 0.1) in the expression of CuSOD mRNA in comparison to that in the diquat group (Figure 2b). Decreased Claudin1 mRNA levels (*p* < 0.05) in the colon were observed in the Cys + diquat, and diquat groups. The mRNA expression of GPX1 and IL-2 were significantly decreased (*p* < 0.05) in the diquat group compared with the control group.

### 3.7. Expression Levels of Cytokines in the Jejunum and Colon

To evaluate the effect of coated cysteamine, the protein expression levels of these cytokines in the jejunum and colon are described in Figure 3. Compared with the diquat group, the protein expression of CuSOD, MnSOD, GPX1, GPX4, ZO-1, Occludin, Claudin1, IL-2, and IL-10 was remarkably reduced (*p* < 0.05) in the Cys + diquat group, whereas the expression of these cytokines was markedly increased in comparison with the control group (Figure 3a). IL-8 protein expression in the jejunum of the control group was higher than that in the diquat group.

The expression levels of the CuSOD, MnSOD, GPX1, GPX4, IL-2, and IL-10 proteins in the Cys + diquat group were lower than those in the diquat group (*p* < 0.05) and higher than those in the control group (*p* < 0.05) (Figure 3b). The corresponding protein levels of IL-8 in the control and Cys + diquat groups were upregulated (*p* < 0.05) compared to the diquat groups.

## 4. Discussion

Cysteamine has activated sulfhydryl and amino groups, which can specifically bind to the disulfide bond of somatostatin (SS), thereby destroying the biological activity of SS, depleting SS in the body, and releasing the inhibitory effect of SS in regulating the body’s anabolism, thus promoting animal growth [23]. A promoting effect on the growth of pigs, finishing pigs and fish has been reported in previous studies [24,25]. Experiments by Du et al. [24] proved that dietary cysteamine supplementation significantly enhanced the feed intake and weight of pigs. Zhu et al. [26] indicated that dietary supplementation with cysteamine-chelated zinc remarkably increased the ADG of pigs and reduced the F/G ratio. When the pigs were added with diquat, we observed negative consequences such as malaise, diarrhea, loss of appetite and decreased activity. In this study, diquat challenge significantly reduced the ADG and ADFI of pigs, and reduced pigs final weight, while the Cys + diquat group showed a significant improvement in this trend, indicating that coated cysteamine addition could alleviate the decline in growth performance caused by diquat.

We speculate that coated cysteamine can alleviate the impairment in growth performance of diquat-induced pigs by improving their antioxidant and anti-inflammatory abilities. Immunoglobulin is an important factor in building the body immunity [27]; IgA eliminates antigen-induced inflammation through specific binding with antigens [28], IgG plays an immune role through antigen-antibody binding [29], and IgM plays an immune response by activating complement proteins in the body [30]. Zhou et al. [31] found that cysteamine supplementation greatly increased IgA, IgG, and IgM in the jejunal mucosa of finishing pigs. Different levels of cysteamine-chelated zinc increased the levels of serum IgA, IgG, and TP in nutrition-restricted pigs but had no significant effect on the level of serum IgM [32]. In the present study, the contents of IgA and IgG were significantly reduced in the diquat group, while the Cys + diquat group exhibited increased levels of IgA, IgG, and IgM, indicating that supplementation with coated cysteamine could promote the synthesis of immunoglobulin, improve the body’s immunity and alleviate the damage caused by diquat.

Liu et al. [33] reported that cysteamine improved pigs growth performance and protein deposition as a feed additive. Growth hormone (GH) can strengthen the intussusception of amino acids (AA), increase the mRNA abundance of AA transporters in the small intestine, and enhance the AA transporter system in the small intestine or cultured primary human trophoblast cells under the interaction of insulin-like growth factor-1 (IGF-1) [34,35]. Dietary supplementation with cysteamine increased the levels of GH and IGF-1 in the body [36]; thus, it could be assumed that the beneficial effect of cysteamine on the AA transporter was achieved by increasing the body’s GH and IGF-1 concentrations. Cysteamine supplementation significantly enhanced the concentrations of cysteine, cystine, and ornithine, and tended to increase the concentrations of essential amino acids (EAA), and Gly [37]. The results of the present study showed that coated cysteamine supplementation resulted in an increase in the contents of Ser and Ile, and reduced the contents of Asp, but the underlying mechanism is still unclear. Therefore, in-depth verification of amino acid transporters should be carried out in the future.

Oxidative stress and inflammation constitute the body’s main defense network, which helps cells survive the stress caused by biochemical, physiological, and pathological stimuli [38,39]. Oxidative stress regulates mitogen-activated protein kinase, extracellular regulatory protein kinase, nuclear factor κ-B (NF-κB), and other signaling pathways by acting on different products, and acts as a “secondary messenger” in the inflammatory response [40]. The main antioxidant enzymes in biological systems such as SOD, glutathione peroxidase (GSH-Px) and CAT can eliminate ROS [41]. The superoxide free radical is converted to H_2_O_2_ by SOD, which is considered to be the first line of defense against cell damage caused by oxygen-free radicals [42]. Diquat treatment reduced the concentrations of serum SOD, CAT, and T-AOC and significantly enhanced the content of MDA [43]. Zhou et al. [31] stated that cysteamine increased the levels of glutathione (GSH) and GSH-Px in pigs, while decreasing the content of MDA. Cysteamine supplementation increased the activity of antioxidant enzymes and the level of GSH to improve the antioxidant status and delay the discoloration of pork [13]. Therefore, it can be demonstrated that cysteamine relieves the oxidative stress state of cells by increasing SOD and GSH in cell tissues and its free sulfhydryl group [44]. Liu et al. [45] found that cysteamine reduced the increase in serum TNF-α, IL-1β, and IL-6 levels caused by *Clostridium perfringens*. The results of the present study showed that coated cysteamine remarkably ameliorated the increase in ALB content and decrease in SOD activity, ALP content, and expression of IL-10 mRNA in the colon caused by diquat. The results indicated that coated cysteamine could enhance immunity and mitigate oxidative stress and inflammation by enhancing the activity of antioxidant enzymes.

The intestinal barrier is a physical barrier composed of a variety of intestinal epithelial cells and cell tight junction complexes, which protect the intestine from damage [46]. The transmembrane proteins Occludin, claudin family, and connexins such as ZO-1, which are important components of tight junctions, are crucial in maintaining intestinal health and integrity [47,48]. In addition, abnormal expression or structural failure of these important proteins damages the intestinal barrier function, resulting in increased intestinal wall permeability [49]. Zhou et al. [31] studied the effect of cysteamine on tight junctions for the first time and showed that cysteamine remarkably increased the expression of Occludin, claudin, and ZO-1 mRNA in the jejunal mucosa, indicating that dietary cysteamine supplementation is conducive to the integrity of the intestinal barrier. Yang et al. [50] showed that cysteamine exacerbated the proliferation of immune cells to boost intestinal mucosal immune functions. When diquat was challenged, it significantly reduced the villus height and the V/C [51], in accordance with the conclusion of this study. At the same time, we found that Occludin, claudin, and ZO-1 mRNA expression in the jejunum of the Cys + diquat group were improved compared to those in the diquat group. This indicates that dietary supplementation with coated cysteamine could improve the intestinal morphology and intestinal barrier function, which reflects the alleviation of inflammation [52].

## 5. Conclusions

This study reported that dietary supplementation with coated cysteamine increased immunoglobulin levels, regulated the mRNA expression of intestinal immune factors to improve jejunal barrier function, and alleviate oxidative stress and inflammation induced by diquat. The above findings provide a theoretical basis for the application of coated cysteamine as a new type of antioxidant feed additive in pigs diets. The effect of coated cysteamine on intestinal microbes and its mechanism needs to be further studied.

## Figures and Tables

**Figure 1 animals-11-02217-f001:**
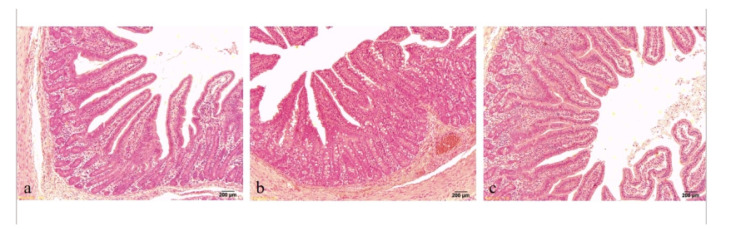
Effects of dietary supplementation of coated cysteamine on jejunal morphology (HE × 40) in pigs induced by diquat. (**a**) Control group; (**b**) Diquat group; (**c**) Cys + diquat group. Scale bar: 200 μm.

**Figure 2 animals-11-02217-f002:**
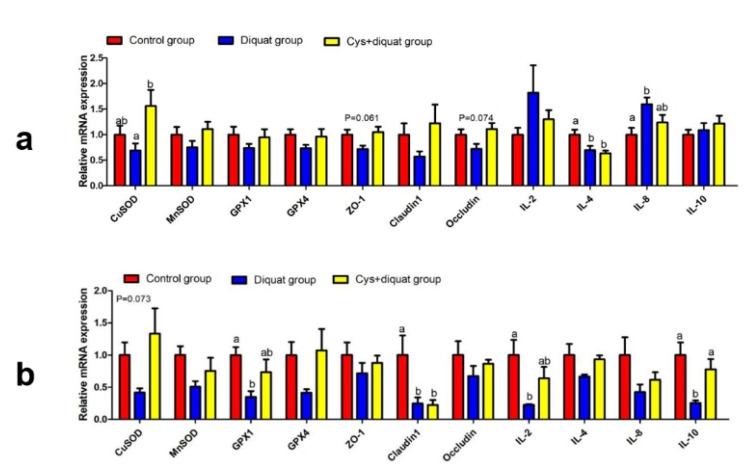
Effects of dietary supplementation of coated cysteamine on the expression of genes related to antioxidant, tight junction and inflammation in pigs (**a**) jejunum and (**b**) colon. CuSOD: Cu-SOD; MnSOD: Mn-SOD; GPX1: glutathione peroxidase 1; GPX4: glutathione peroxidase 4; ZO-1: zonula occludens-1; IL-2: interleukin-2; IL-4: interleukin-4; IL-8: interleukin-8; IL-10: interleukin-10. Results are expressed as means ± SEM (n = 8). Means in the same row with different letters were significantly different (*p* < 0.05).

**Figure 3 animals-11-02217-f003:**
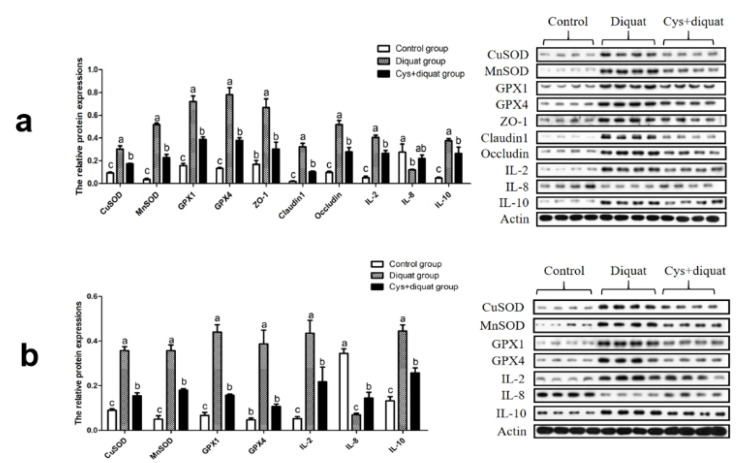
Effects of dietary supplementation of coated cysteamine on the expression of antioxidant-, tight junction- and inflammation-related proteins in pigs (**a**) jejunum and (**b**) colon. CuSOD: Cu-SOD; MnSOD: Mn-SOD; GPX1: glutathione peroxidase 1; GPX4: glutathione peroxidase 4; ZO-1: zonula occludens-1; IL-2: interleukin-2; IL-4: interleukin-4; IL-8: interleukin-8; IL-10: interleukin-10. Results are expressed as means ± SEM (n = 8). Means in the same row with different letters were significantly different (*p* < 0.05).

**Table 1 animals-11-02217-t001:** Basic diet composition and nutritional level of pigs.

Ingredient, g/kg	
Corn	300.00
Extrude corn	210.00
Soybean meal	160.00
Extrude soybean	120.00
Fish meal	30.00
Wheat middling and red dog	28.50
Broken rice	50.00
Soybean oil	20.00
Sugar	20.00
Glu	20.00
Organic acid calcium	6.00
Calcium phosphate	12.00
L-Lysine-HCl	6.40
DL-Met	0.80
Thr	1.20
Mineral premix ^1^	1.50
Vitamin premix ^1^	0.40
Acidifier	5.00
Zinc oxide	3.20
Limestone	5.00
**Nutrient component, %**	
Digestible energy kcal/kg	3464.00
Crude protein	19.25
Lys	1.52
Met	0.40
Thr	0.86
Try	0.30

^1^ Providing the following amounts of vitamins and minerals per kilogram on an as-fed basis: Zn (ZnO) 50 mg; Cu (CuSO_4_) 20 mg; Mn (MnO) 55 mg; Fe (FeSO_4_) 100 mg; I (KI) 1 mg; Co (CoSO_4_) 2 mg; Se (Na_2_SeO_3_) 0.3 mg; vitamin A, 8255 IU; vitamin D3, 2000 IU; vitamin E, 40 IU; vitamin B1, 2 mg; vitamin B2, 4 mg; pantothenic acid, 15 mg; vitamin B6, 10 mg; vitamin B12, 0.05 mg; nicotinic acid, 30 mg; folic acid, 2 mg; vitamin K3, 1.5 mg; biotin, 0.2 mg; choline chloride, 800 mg; vitamin C, 100 mg.

**Table 2 animals-11-02217-t002:** The sequences of the primers used for quantitative real-time polymerase chain reaction (PCR).

Gene	Accession No.	Primer 5′–3′	Size (bp)	Tm °C
CuSOD	NM_001190422.1	F:CAGGTCCTCACTTCAATCC	255	54.89
		R:CCAAACGACTTCCACCAT		
MnSOD	NM_214127.2	F:GGACAAATCTGAGCCCTAACG	159	58.72
		R:CCTTGTTGAAACCGAGCC		
GPX1	NM_214201.1	F:TGGGGAGATCCTGAATTG	184	53.05
		R:GATAAACTTGGGGTCGGT		
GPX4	NM_214407.1	F:GATTCTGGCCTTCCCTTGC	173	58.5
		R:TCCCCTTGGGCTGGACTTT		
ZO-1	XM_021098827.1	F:CCTGCTTCTCCAAAAACTCTT	252	56.35
		R:TTCTATGGAGCTCAACACCC		
Claudin1	NM_001244539.1	F:AAGGACAAAACCGTGTGGGA	247	59.74
		R:CTCTCCCCACATTCGAGATGATT		
Occludin	NM_001163647.2	F:ACGAGCTGGAGGAAGACTGGATC	238	63.63
		R:CCCTTAACTTGCTTCAGTCTATTG		
IL-2	NM_213861.1	F:TGCACTAACCCTTGCACTCA	100	59.53
		R:CAACTGTAAATCCAGCAGCAA		
IL-4	NM_214123.1	F:CCCAACTGATCCCAACCCTG	139	60.32
		R:AGCTCCATGCACGAGTTCTT		
IL-8	NM_213867.1	F:TGAGAAGCAACAACAACAGCA	129	58.91
		R:CAGCACAGGAATGAGGCATA		
IL-10	NM_214041.1	F:GGGCTATTTGTCCTGACTGC	105	58.62
		R:GGGCTCCCTAGTTTCTCTTCC		

Abbreviations used: CuSOD: Cu-SOD; MnSOD: Mn-SOD; GPX1: glutathione peroxidase 1; GPX4: glutathione peroxidase 4; ZO-1: zonula occludens-1; IL-2: interleukin-2; IL-4: interleukin-4; IL-8: interleukin-8; IL-10: interleukin-10.

**Table 3 animals-11-02217-t003:** Effects of dietary supplementation of coated cysteamine on growth performance of pigs induced by diquat.

Items	Control	Diquat	Cys + Diquat	*p* Value
Initial weight kg	9.04 ± 0.50	9.04 ± 0.43	9.16 ± 0.23	0.819
Final weight kg	16.64 ± 0.78 ^a^	14.54 ± 1.18 ^b^	15.83 ± 1.38 ^a^	0.010
1–7 days				
ADG kg/d	0.42 ± 0.11	0.37 ± 0.05	0.40 ± 0.06	0.420
ADFI kg/d	0.70 ± 0.08	0.74 ± 0.07	0.72 ± 0.06	0.842
F/G	1.71 ± 0.26	1.92 ± 0.29	1.88 ± 0.16	0.164
8–14 days				
ADG kg/d	0.36 ± 0.04	0.37 ± 0.05	0.33 ± 0.05	0.822
ADFI kg/d	0.71 ± 0.06	0.73 ± 0.06	0.70 ± 0.09	0.411
F/G	1.93 ± 0.20	2.00 ± 0.24	2.08 ± 0.65	0.113
15–21 days				
ADG kg/d	0.36 ± 0.05 ^a^	0.16 ± 0.15 ^b^	0.24 ± 0.04 ^b^	0.040
ADFI kg/d	0.59 ± 0.04 ^a^	0.45 ± 0.16 ^b^	0.52 ± 0.06 ^ab^	0.022
F/G	1.49 ± 0.52	2.16 ± 0.64	1.76 ± 1.36	0.371

Note: ADG: average daily gain; ADFI: average daily feed intake; F/G: feed-to-gain ratio. Results are expressed as means ± SEM (n = 8). Means in the same row with different letters were significantly different (*p* < 0.05).

**Table 4 animals-11-02217-t004:** Effects of dietary supplementation of coated cysteamine on serum biochemical indexes of pigs induced by diquat.

Items	Control	Diquat	Cys + Diquat	*p* Value
TP g/L	61.92 ± 1.91 ^a^	62.14 ± 1.45 ^a^	55.47 ± 2.98 ^b^	<0.001
ALB g/L	32.00 ± 2.00 ^b^	36.98 ± 4.89 ^a^	28.68 ± 1.85 ^b^	0.002
ALT U/L	61.15 ± 15.21 ^a^	35.86 ± 6.44 ^b^	61.97 ± 2.48 ^a^	0.001
AST U/L	67.67 ± 13.65	63.20 ± 32.71	57.83 ± 9.41	0.704
ALP U/L	495.83 ± 40.45 ^a^	319.20 ± 65.98 ^b^	470.33 ± 163.18 ^a^	0.036
CHOL mmol/L	2.91 ± 0.22	2.94 ± 0.40	2.49 ± 0.49	0.118
TG mmol/L	0.64 ± 0.12	0.62 ± 0.22	0.57 ± 0.12	0.711
GLU mmol/L	3.90 ± 1.38 ^b^	5.62 ± 0.40 ^a^	3.63 ± 1.03 ^b^	0.016
BUN mmol/L	3.13 ± 0.96	3.26 ± 0.50	3.23 ± 0.48	0.949
IgA μg/mL	796.84 ± 22.31 ^a^	514.42 ± 161.75 ^b^	583.16 ± 118.98 ^b^	0.033
IgG μg/mL	8437.97 ± 943.92 ^a^	6158.59 ± 907.82 ^b^	7697.12 ± 738.59 ^a^	0.001
IgM μg/mL	489.73 ± 107.78	396.12 ± 104.85	449.36 ± 90.26	0.304

Note: TP: total protein; ALB: albumin; ALT: alanine aminotransferase; AST: aspartate aminotransferase; ALP: alkaline phosphatase; CHOL: cholesterol; TG: triglycerides; GLU: glucose; BUN: blood urea nitrogen; IgA: immunoglobulin A; IgG: immunoglobulin G; IgM: immunoglobulin M. Results are expressed as means ± SEM (n = 8). Means in the same row with different letters were significantly different (*p* < 0.05).

**Table 5 animals-11-02217-t005:** Effect of dietary supplementation of coated cysteamine on serum amino acids in pigs induced by diquat.

Items (μg/mL)	Control	Diquat	Cys + Diquat	*p* Value
Essential Amino				
Lys	12.61 ± 2.34	10.78 ± 1.69	11.04 ± 0.97	0.225
Phe	4.14 ± 0.51	3.69 ± 0.42	3.43 ± 0.65	0.115
Thr	5.72 ± 1.00	5.00 ± 1.66	5.14 ± 0.96	0.591
Val	9.10 ± 1.72	8.24 ± 1.29	7.14 ± 1.49	0.145
Met	1.29 ± 0.32	1.04 ± 0.19	1.03 ± 0.33	0.274
His	1.00 ± 0.25	1.27 ± 0.37	0.80 ± 0.23	0.065
Ile	6.01 ± 1.27 ^a^	4.39 ± 0.65 ^b^	5.83 ± 0.99 ^a^	0.046
Leu	8.50 ± 1.71	7.83 ± 0.87	7.62 ± 1.71	0.607
Tyr	4.58 ± 0.65	4.19 ± 0.88	4.39 ± 0.87	0.730
Non-essential Amino				
Glu	12.82 ± 4.02	14.91 ± 2.38	11.78 ± 1.54	0.272
Arg	8.70 ± 2.18	9.43 ± 2.31	8.28 ± 1.68	0.684
Ser	7.35 ± 0.97 ^a^	5.61 ± 0.49 ^b^	7.47 ± 0.48 ^a^	0.002
Asp	0.70 ± 0.17 ^a^	1.03 ± 0.38 ^b^	0.62 ± 0.14 ^a^	0.045
Ala	17.97 ± 3.48	21.30 ± 2.61	18.27 ± 2.89	0.191
Gly	41.26 ± 5.51 ^a^	37.09 ± 4.90 ^ab^	32.37 ± 4.93 ^b^	0.043
Cys	3.84 ± 0.55	3.46 ± 0.56	3.25 ± 0.77	0.314
Pro	8.29 ± 0.89	8.26 ± 0.39	7.76 ± 0.25	0.320

Note: Lys: lysine; Phe: phenylalanine; Thr: threonine; Val: valine; Met: methionine; His: histidine; Ile: isoleucine; Leu: leucine; Tyr: tyrosine; Glu: glutamic acid; Arg: arginine; Ser: serine; Asp: aspartic acid; Ala: alanine; Gly: glycine; Cys: cysteine; Pro: proline. Results are expressed as means ± SEM (n = 8). Means in the same row with different letters were significantly different (*p* < 0.05).

**Table 6 animals-11-02217-t006:** Effects of dietary supplementation of coated cysteamine on jejunal morphology of pigs induced by diquat.

Items	Control	Diquat	Cys + Diquat	*p*-Value
Villous height, μm	458.99 ± 32.02 ^a^	362.99 ± 66.59 ^b^	424.23 ± 66.36 ^ab^	0.060
Crypt depth, μm	202.09 ± 17.79	206.93 ± 32.97	200.66 ± 13.77	0.904
Villous height/Crypt depth	2.43 ± 0.27 ^a^	2.02 ± 0.25 ^b^	2.12 ± 0.23 ^ab^	0.056

Note: Results are expressed as means ± SEM (n = 8). Means in the same row with different letters were significantly different (*p* < 0.05).

**Table 7 animals-11-02217-t007:** Effect of dietary supplementation of coated cysteamine on the serum antioxidant indexes of pigs induced by diquat.

Items	Control	Diquat	Cys + Diquat	*p*-Value
MPO ng/g	1573.89 ± 799.26	1773.90 ± 566.79	1044.81 ± 112.82	0.168
CAT U/mg prot	0.13 ± 0.11	0.11 ± 0.09	0.18 ± 0.02	0.503
SOD U/g prot	12.46 ± 7.69 ^b^	11.17 ± 7.47 ^b^	37.35 ± 22.28 ^a^	0.016
MDA nmol/g port	3.14 ± 3.22	5.95 ± 4.33	2.82 ± 2.01	0.287
T-AOC U/g	0.63 ± 0.16	0.88 ± 0.25	0.75 ± 0.09	0.112

Note: MPO: myeloperoxidase; CAT: catalase; SOD: superoxide dismutase; MDA: malondialdehyde; T-AOC: total antioxidant capacity. Results are expressed as means ± SEM (n = 8). Means in the same row with different letters were significantly different (*p* < 0.05).

## Data Availability

None of the data were deposited in an official repository.

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
