# Peer review of "Effects of Coated Cysteamine on Oxidative Stress and Inflammation in Weaned Pigs"

_animals, 2021, doi:10.3390/ani11082217_

Round 1

Reviewer 1 Report

This is a study by Wang and colleagues, examining the effects of coated cysteamine against diquat-induced oxidative stress and inflammation. They found that the cysteamine supplementation protects against epithelial damage, increases intestinal barrier function and enhances antioxidant enzyme levels. Overall, the study is potentially interesting and focuses on a relevant subject aiming at the understanding of beneficial effects of coated cysteamine in preventing intestinal damages caused by oxidative stress. There are few grammatical errors, corrections and clarifications to be addressed.

Specific points:

Lines 16-18: “the total prohibition of antibiotics has made it necessary …” there is no discussion about antibiotics and how it is related to oxidative stress. Please add the justification or rewrite this sentence.

Lines 22-23: Please rewrite this sentence.

Lines 94-94: Do you mean injections of diquat for 7 consecutive days? There is no recovery period? Please make it clear.  

Lines 98-100: “cysteamine was added to the feed in the form of coated cysteamine, which contains 27% cysteamine hydrochloride…”  80 mg/kg cysteamine diet is the coated form or cysteamine hydrochloride?

Lines 217-218: In fact, the differences between the groups are rather small from the figures. This might be particularly relevant considering the dose of cysteamine applied in this study. How did the authors determine such concentration? Was there a pilot study before the current study?

From previous studies, diquat can cause damages especially in liver and small intestine. Here you also measured the inflammation levels and barrier function in colon. Did you see any change in the histology of the colon?

Author Response

Dear reviewers,

Thank you very much for your letter and the comments on our manuscript. Now we have revised the manuscript according to your comments. All revisions were marked red in the manuscript.

Response to Reviewer 1 Comments

Point 1: Lines 16-18: “the total prohibition of antibiotics has made it necessary …” there is no discussion about antibiotics and how it is related to oxidative stress. Please add the justification or rewrite this sentence.

Response: Line 16-18: “the total prohibition of antibiotics has made it necessary …” was revised as “Finding an effective way to deal with oxidative stress and inflammation is an urgent problem for the modern pig industry.”

Point 2: Lines 22-23: Please rewrite this sentence.

Response: Lines 22: “In this study, it is proved that the addition of coated cysteamine can enhance the level of immunoglobulin and the expression of immune factors, and improve the production performance by improving the intestine barrier function and immune function, thereby alleviating the oxidative stress and Inflammation caused by diquat.” was revised as “In this study, we found that dietary supplementation of coated cysteamine could enhance the level of immunoglobulin and the expression of intestinal immune factors in the weaned pigs induced by diquat. It is proved that dietary supplementation with coated cysteamine can improve the intestine barrier function and immune function.”

Point 3: Lines 94-94: Do you mean injections of diquat for 7 consecutive days? There is no recovery period? Please make it clear. 

Response: Lines 97: We injected diquat continuously for 7 days from the 14th day without any recovery period.

Point 4: Lines 98-100: “cysteamine was added to the feed in the form of coated cysteamine, which contains 27% cysteamine hydrochloride…”  80 mg/kg cysteamine diet is the coated form or cysteamine hydrochloride?

Response: We added 80mg/kg coated cysteamine to the basal diet. Cysteamine hydrochloride is the main component of coated cysteamine, accounting for 27% of its content.

Point 5: Lines 217-218: In fact, the differences between the groups are rather small from the figures. This might be particularly relevant considering the dose of cysteamine applied in this study. How did the authors determine such concentration? Was there a pilot study before the current study?

Response: The previous study showed that dietary supplementation of 70mg/kg coated cysteamine had the best antioxidant effect on finishing pigs (Bai et al., 2017). The experiment was performed by our lab. Therefore, based on the studies of Bai and others, we finally decided to set the supplemental level of coated cysteamine as 80mg/kg.

Point 6: From previous studies, diquat can cause damages especially in liver and small intestine. Here you also measured the inflammation levels and barrier function in colon. Did you see any change in the histology of the colon?

Response: We focused more on detecting morphological and histological changes in the small intestine, so we did not detect histological changes in the colon. However, according to the results of mRNA and protein expression of colon-related factors, it can be inferred that diquat caused some damage to the colon, which was alleviated by the supplementation of coated cysteamine.

Thanks again for your letter and the comments on our manuscript.

Best wishes,

Sincerely yours,

Shanshan Wang

Reviewer 2 Report

Mayor comments:

Line 86: Please describe the negative consequences on animals welfare of your 2nd treatment where diquat was added

Line 114: if the diquat administration was performed on day 14th you probably need to weight animals weekly to approach better to possible group performance differences. The first 14 days only compare control v/s control v/s cysteamine and this could confuse the interpretation of results. Why did you measure feed take daily? Please describe in M&M the feeders that you used, and the pen distribution and allocation of animals. Animals were housed in groups or alone? 

Results and discussion: probably a positive control “cysteamine” is essential to discuss the real effects of its inclusion in nursery pigs. Author could also discuss with performance results before the administration of diquat

Minor comments:

Simple Summary: Please add the species

Line 29 and elsewhere: piglets è pigs

Keywords: Alphabetic order

Line 91: Please describe in detail “diquat”

Line 99: Please describe in detail the coated process

Table 1: It is not clear if numbers are in percent or in g/Kg. Please change the Nutritional composition to the bottom of the table

Line 176: Please describe the dependent variables that you analysed (and when) and the independent variable (Group).

Author Response

Dear reviewers,

Thank you very much for your letter and the comments on our manuscript. Now we have revised the manuscript according to your comments. All revisions were marked red in the manuscript.

Response to Reviewer 2 Comments

Point 1: Line 86: Please describe the negative consequences on animals welfare of your 2nd treatment where diquat was added.

Response: I have added “When the piglets were added with diquat, we observed negative consequences such as malaise, diarrhea, loss of appetite, and decreased activity.” to Line 88.

Point 2: Line 114: if the diquat administration was performed on day 14th you probably need to weight animals weekly to approach better to possible group performance differences. The first 14 days only compare control v/s control v/s cysteamine and this could confuse the interpretation of results. Why did you measure feed take daily? Please describe in M&M the feeders that you used, and the pen distribution and allocation of animals. Animals were housed in groups or alone?

Response: I added data in Table3 on Line195. I have added "Our pigs are fed in single pens" to Line 104.

Point 3: Results and discussion: probably a positive control “cysteamine” is essential to discuss the real effects of its inclusion in nursery pigs. Author could also discuss with performance results before the administration of diquat.

Response: I have revised the description of the “results and discussion” in the paper.

Point 4: Simple Summary: Please add the species

Response: Line 19: “….thereby alleviating the oxidative stress and Inflammation caused by diquat.” was revised as “In this study, we found that dietary supplementation of coated cysteamine could enhance the level of immunoglobulin and the expression of intestinal immune factors in the weaned pigs induced by diquat.”

Point 5: Line 29 and elsewhere: piglets è pigs

Response: All “piglets” has been revised as “pigs”.

Point 6: Keywords: Alphabetic order

Response: Line 42: “Keywords: cysteamine; oxidative stress; intestinal barrier; inflammatory cytokines” was revised as “Keywords: cysteamine; inflammatory cytokines; intestinal barrier; oxidative stress”

Point 7: Line 91: Please describe in detail “diquat”

Response: Line 97: “On the 14th day, the piglets were weighed and treated with diquat.” Was revised as “On the 14th day, the piglets were weighed and treated with diquat for 7d (Sigma-Aldrich, St. Louis, USA).

Point 8: Line 99: Please describe in detail the coated process

Response: This experiment uses a processed coated cysteamine product. Coated cysteamine is a feed stable cysteamine hydrochloride produced by advanced microencapsulation technology. It adopts advanced microcapsule coating technology and special coating wall materials, with an encapsulation rate of 100%. The coated cysteamine have better fluidity and stability, and better sustained-release performance, which effectively overcomes the disadvantages of easy oxidation and moisture absorption of cysteamine hydrochloride. It has successfully avoided the impact of the strong ammonia odor of cysteamine exposed amino groups on the feed intake of livestock and poultry, and ensured that it can be effectively released in the digestive tract of livestock and poultry.

Point 9: Table 1: It is not clear if numbers are in percent or in g/Kg. Please change the Nutritional composition to the bottom of the table

Response: I have revised Table 1 of this article in Line 106.

Point 10: Line 176: Please describe the dependent variables that you analysed (and when) and the independent variable (Group).

Response: I've added “The piglets were treated with independent variables cysteamine and diquat, and explored their effects on various indicators.” to Line 182.

Thanks again for your letter and the comments on our manuscript.

Best wishes,

Sincerely yours,

Shanshan Wang

Reviewer 3 Report

Dear Authors:

The research work and data are passable, but the MS is unacceptable as the citations are totally incomplete. This needs to be corrected.

This is a MS that using standard clinical type assessments for oxidative/inflammation status (except PCR) and in many ways is another product testing experiment.

The real problem here is the MS preparation especially the Lit Cited section. Almost all citation LACK the identification of the journals; there are also examples such as ref 16. These must all be corrected until the MS can be judged properly. Look at ref 48 as example..we have title, date and volume, but no journal and no pages. I wonder if this was a translation issues.

In any case as presently presented, this MS is totally unacceptable!!

The MS must be reworked especially the Citation section. 

Author Response

Dear reviewers,

Thank you very much for your letter and the comments on our manuscript. Now we have revised the manuscript according to your comments. All revisions were marked red in the manuscript.

Response to Reviewer 3 Comments

Point 1: Dear Authors:

The research work and data are passable, but the MS is unacceptable as the citations are totally incomplete. This needs to be corrected.

This is a MS that using standard clinical type assessments for oxidative/inflammation status (except PCR) and in many ways is another product testing experiment.

The real problem here is the MS preparation especially the Lit Cited section. Almost all citation LACK the identification of the journals; there are also examples such as ref 16. These must all be corrected until the MS can be judged properly. Look at ref 48 as example..we have title, date and volume, but no journal and no pages. I wonder if this was a translation issues.

In any case as presently presented, this MS is totally unacceptable!!

The MS must be reworked especially the Citation section.

Response: I have revised the format of the reference.

Thanks again for your letter and the comments on our manuscript.

Best wishes,

Sincerely yours,

Shanshan Wang

Round 2

Reviewer 2 Report

Line 86: Please describe the negative consequences on animals welfare of your 2nd treatment where diquat was added

Authors added information about diquat consequences. However, probably this is not the best place to include this information. Move these lines to other place as the beginning of the results section. Moreover did you register health and behaviour of animals with a supervision guideline?

Line 114: if the diquat administration was performed on day 14th you probably need to weight animals weekly to approach better to possible group performance differences. The first 14 days only compare control v/s control v/s cysteamine and this could confuse the interpretation of results. Why did you measure feed take daily? Please describe in M&M the feeders that you used, and the pen distribution and allocation of animals. Animals were housed in groups or alone? 

Authors included information of performance on table 3. However you need to improve this table. Please add animals body weigh at day 7, 14, 21 and 35. What does it mean 1W 2 W and 2W? 1-7 days 8-14 days and 15-21 days? The ADG of control group during 3W is lower than ADg during 1W, it is possible? Please describe in M&M the feeders and drinkers that you used. If animals were housed alone how did you enrich their environment?

Results and discussion: probably a positive control “cysteamine” is essential to discuss the real effects of its inclusion in nursery pigs. Author could also discuss with performance results before the administration of diquat

If no differences were found between the control groups and the cysteamine at the beginning of the experiment (before the diquat administration) why producers need to use cysteamine? The discussion was not modified as I recommended. 

Minor comments:

Simple Summary: Please add the species

Ok

Line 29 and elsewhere: piglets è pigs

Ok

Keywords: Alphabetic order

Ok

Line 91: Please describe in detail “diquat”

More details are needed

Line 99: Please describe in detail the coated process

Please include this information into the text

Table 1: It is not clear if numbers are in percent or in g/Kg. Please change the Nutritional composition to the bottom of the table

Still not clear

Line 176: Please describe the dependent variables that you analysed (and when) and the independent variable (Group).

Ok

Author Response

Dear reviewers,

Thank you very much for your letter and the comments on our manuscript. Now we have revised the manuscript according to your comments. All revisions were marked red in the manuscript.

Response to Reviewer Comments

Point 1: Line 86: Please describe the negative consequences on animals welfare of your 2nd treatment where diquat was added

Authors added information about diquat consequences. However, probably this is not the best place to include this information. Move these lines to other place as the beginning of the results section. Moreover did you register health and behaviour of animals with a supervision guideline?

Response: I have shown “When the pigs were added with diquat, we observed negative consequences such as malaise, diarrhea, loss of appetite, and decreased activity.” move to Line 311. And I have added “All animals used in this study were humanely managed according to the Chinese Guidelines for Animal Welfare.” to Line 92.

Point 2: Line 114: if the diquat administration was performed on day 14th you probably need to weight animals weekly to approach better to possible group performance differences. The first 14 days only compare control v/s control v/s cysteamine and this could confuse the interpretation of results. Why did you measure feed take daily? Please describe in M&M the feeders that you used, and the pen distribution and allocation of animals. Animals were housed in groups or alone?

Authors included information of performance on table 3. However you need to improve this table. Please add animals body weigh at day 7, 14, 21 and 35. What does it mean 1W 2 W and 2W? 1-7 days 8-14 days and 15-21 days? The ADG of control group during 3W is lor than ADg during 1W, it is possible? Please describe in M&M the feeders and drinkers that you used. If animals were housed alone how did you enrich their environment?

Response: In Table3, “1w, 2w and 3w” have been revised as “1-7 days, 8-14 days, and 15-21 days” respectively. We do not have weekly weighing in order to minimize the stress to the animals and close to the actual production situation. I think it may be the individual differences among pigs that cause the ADG of 3w to be lower than that of 1w. I have added “Before the test, the piggery was cleaned and disinfected thoroughly according to the piggery management process. During the test, keep the house clean and dry, ensure air circulation, and clean the house every day. All piglets are housed in an environmentally well nursery facility with good heat preservation facilities and a mechanical ventilation system. The adjacent fields are separated by steel tubes to ensure that the piglets are not completely isolated from each other. The piggery is equipped with an automatic drinking water device, manual feeding is adopted and the trough is cleaned in time.” to Line 108. And I have modified the data in Table3.

Point 3: Results and discussion: probably a positive control “cysteamine” is essential to discuss the real effects of its inclusion in nursery pigs. Author could also discuss with performance results before the administration of diquat

If no differences were found between the control groups and the cysteamine at the beginning of the experiment (before the diquat administration) why producers need to use cysteamine? The discussion was not modified as I recommended.

Response: Growth performance is an indicator of a comprehensive reaction in many aspects. The addition of cysteamine in this experiment shows a little effect on antioxidant, and may not have a significant effect on growth performance. I have revised the part of the discussion.

Point 4: Line 91: Please describe in detail “diquat”

More details are needed

Response: Line 102: “On the 14th day, the pigs were weighed and treated with diquat for 7d (Sigma-Aldrich, St. Louis, USA).” was revised as “On the 14th day, the pigs were weighed and treated with diquat for 7d (85-00-7, >95.00%, Sigma-Aldrich, St. Louis, USA).”

Point 5: Line 99: Please describe in detail the coated process

Please include this information into the text

Response: I have added “Coated cysteamine is a feed stable cysteamine hydrochloride produced by advanced microencapsulation technology. It adopts advanced microcapsule coating technology and special coating wall materials, with an encapsulation rate of 100%. The coated cysteamine has better fluidity and stability, better sustained-release performance, and better tolerance.” to Line 76. More detailed content is suspected of trade secrets, inconvenient to introduce too much.

Point 6: Table 1: It is not clear if numbers are in percent or in g/Kg. Please change the Nutritional composition to the bottom of the table

Still not clear

Response: I have modified Table1.

Thanks again for your letter and the comments on our manuscript.

Best wishes,

Sincerely yours,

Shanshan Wang